# A bacterial checkpoint protein for ribosome assembly moonlights as an essential metabolite-proofreading enzyme

Ankita J. Sachla[1] & John D. Helmann[1]

In eukaryotes, adventitious oxidation of erythrose-4-phosphate, an intermediate of the pentose phosphate pathway (PPP), generates 4-phosphoerythronate (4PE), which inhibits 6-phosphogluconate dehydrogenase. 4PE is detoxified by metabolite-proofreading phosphatases such as yeast Pho13. Here, we report that a similar function is carried out in *Bacillus subtilis* by CpgA, a checkpoint protein known to be important for ribosome assembly, cell morphology and resistance to cell wall-targeting antibiotics. We find that Δ*cpgA* cells are intoxicated by glucose or other carbon sources that feed into the PPP, and that CpgA has high phosphatase activity with 4PE. Inhibition of 6-phosphogluconate dehydrogenase (GndA) leads to intoxication by 6-phosphogluconate, a potent inhibitor of phosphoglucose isomerase (PGI). The coordinated shutdown of PPP and glycolysis leads to metabolic gridlock. Overexpression of GndA, PGI, or yeast Pho13 suppresses glucose intoxication of Δ*cpgA* cells, but not cold sensitivity, a phenotype associated with ribosome assembly defects. Our results suggest that CpgA is a multifunctional protein, with genetically separable roles in ribosome assembly and metabolite proofreading.

---

[1] Department of Microbiology, Cornell University, 370 Wing Hall, 123 Wing Drive, Ithaca, NY 14853-8101, USA. Correspondence and requests for materials should be addressed to J.D.H. (email: jdh9@cornell.edu)

The ribosome is an abundant and exceptionally complex structure whose assembly and function dominates bacterial physiology. During rapid growth of bacteria such as *Bacillus subtilis*, the majority of RNA polymerase is engaged in the synthesis of ribosomal RNA (rRNA). Ribosomes comprise up to 50% of cell mass, and translation consumes up to 2/3 of cellular energy[1]. As expected for such an energetically demanding process, ribosome synthesis and assembly is highly regulated, and defects can impose severe fitness costs on cells.

Ribosome assembly requires the efficient processing of the precursor rRNA transcripts followed by the ordered assembly of ribosomal proteins to generate the 30S and 50S subunits. Assembly of the ribosome occurs rapidly in vivo, in a process facilitated by RNA helicases, rRNA modification enzymes, and ribosome-assembly GTPases (RA-GTPases)[2]. The RA-GTPases are universally conserved proteins that couple GTP hydrolysis to specific checkpoints in assembly. In *B. subtilis*, the six RA-GTPases include RbgA, YphC, and YsxC implicated in assembly of the 50S subunit[3,4], and Era, YqeH, and CpgA to facilitate assembly of the 30S subunit[2,5,6]. Of these six RA-GTPases, four are essential and mutants in *yqeH* and *cpgA* are growth impaired[5,7–9]. CpgA (circularly-permuted GTPase) is also important for normal cell morphology, proper deposition of the peptidoglycan sacculus, and intrinsic resistance to antibiotics affecting both the ribosome and cell wall synthesis[5,10]. Whether or not these various phenotypes are related to the ribosome-assembly role is not resolved.

CpgA is also a target for PrkC, a eukaryotic-like Hanks Ser/Thr kinase with surface-exposed penicillin binding protein and Ser/Thr kinase associated (PASTA) domains implicated in muropeptide sensing[11–13]. Indeed, *cpgA* is co-transcribed with *prkC* and *prpC*, encoding the cognate phosphatase for PrkC[14]. Using phosphomimetic and phosphoablative variants of CpgA, it has been proposed that phosphorylation of CpgA at Thr166 increases intrinsic GTPase activity, enhances association with 30S ribosomal subunits to aid in ribosome maturation, and is necessary for normal cell morphology[6]. However, whether CpgA phosphorylation affects its activities related to peptidoglycan deposition and antibiotic sensitivity is not clear.

Here, we report studies in which we unexpectedly discovered that CpgA functions as a broad specificity phosphatase that protects cells against the deleterious effects of underground metabolism. A Δ*cpgA* mutant is sensitive to β-lactam antibiotics and is unable to grow in the presence of glucose or gluconate. These metabolic restrictions arise from the inhibition of 6-phosphogluconate dehydrogenase (GndA) by 4-phosphoerythronate (4PE), an adventitious oxidation product arising from the pentose phosphate pathway (PPP) intermediate erythrose-4-phosphate. 4PE initiates an inhibition cascade in which 6-phosphogluconate accumulates and competitively inhibits phosphoglucose isomerase (PGI), leading to metabolic gridlock (Fig. 1), bacteriostasis, and ultimately cell death. CpgA prevents metabolic intoxication by cleansing the metabolite pool of potentially toxic molecules, including 4PE. These results document an unexpected link between CpgA and carbon catabolism that is important in preventing deleterious effects arising from enzyme promiscuity on cell physiology.

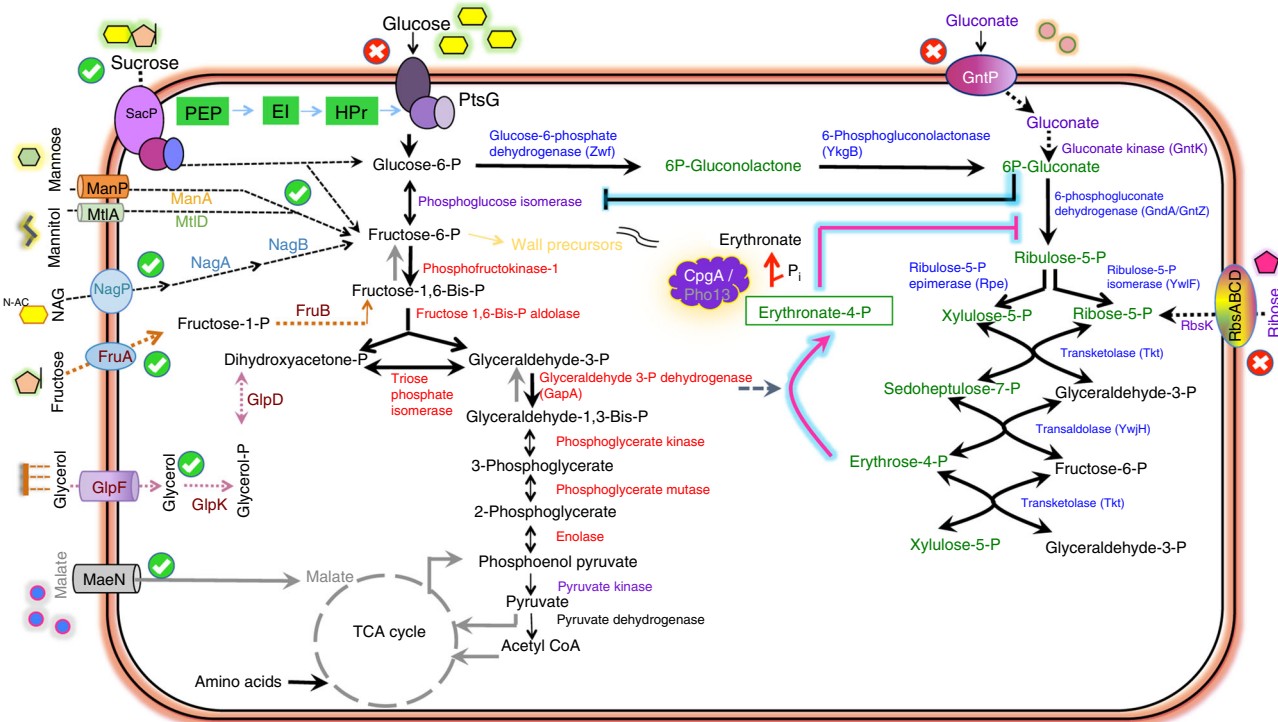

**Fig. 1** CpgA is pivotal in prevention of 4PE-mediated metabolic gridlock. PtsG functions as a glucose PTS importer (EII[Glc] class) working with HPr (Histidine protein), EI (Enzyme I), and PEP (phosphoenolpyruvate) to import glucose (yellow hexagons). Other enzymes and proteins mentioned in the text are shown. Metabolites of glycolysis are depicted in black (enzymes in red), and pentose phosphate pathway intermediates are depicted in green (enzymes in blue). Highlighted arrows illustrate the accidental metabolite 4PE arising from the promiscuous reaction of GapA with erythrose-4-phosphate to generate 4-P-erythronate (4PE), which inhibits GndA/GntZ. CpgA or yeast Pho13 (depicted as purple protein) convert 4PE to erythronate. Inhibition of GndA/GntZ causes accumulation of 6-phosphogluconate which further blocks phosphoglucose isomerase (PGI) leading to a metabolic gridlock in core pathways. A green circle with check mark indicates carbon sources that support growth of the *cpgA* null mutant, and those carbon sources that lead to intoxication are shown as red circles with cross marks

## Results

**A ΔcpgA mutant is intoxicated by glucose or gluconate**. Several substrates of the PrkC kinase/PrpC phosphatase system have been linked to cell wall assembly and intrinsic antibiotic resistance, including WalR, GlmR/YvcK, GpsB, and CpgA[5,6,15–20]. The *prpC-prkC-cpgA* operon is conserved in the *Firmicutes* (Fig. 2a), and CpgA is a target for PrkC phosphorylation[6,18], suggesting that these proteins may function in the same pathway. We therefore assessed the role of these genes in intrinsic resistance to cefuroxime (CEF), a β-lactam antibiotic that serves as a sensitive indicator of cell wall assembly defects in *B. subtilis*. We used the BKE/BKK collection of gene disruptants[21] to generate in-frame, clean deletions in *prkC* and *prpC*. The resulting Δ*prkC* strain was slightly more sensitive to CEF than WT, whereas the Δ*prpC* strain was slightly more resistant (Fig. 2a). The Δ*cpgA* mutant was much more sensitive than Δ*prkC*, indicating that CpgA has functions relevant to cell wall homeostasis independent of PrkC. In liquid LB medium, we noted a 10-fold increase in the sub-lethal concentration of CEF for Δ*cpgA* strains (~0.01 μg ml$^{-1}$) as compared with WT (~0.1 μg ml$^{-1}$) (Supplementary Fig. 1a and b). Moreover, in the Δ*cpgA* background, mutation of either *prkC* or *prpC* no longer had an observable effect on CEF sensitivity (Fig. 2a). Based on these findings, we conclude that CpgA plays a dominant role in intrinsic CEF resistance, and the more modest effects noted for the PrkC/PrpC system may be due to modification of CpgA.

We used *mariner* transposon (mTn) mutagenesis to select for colonies with increased CEF resistance in the Δ*cpgA* parent strain on LB medium. The mTn insertion sites were mapped by sequencing to within or near genes encoding an alternate lipotechoic acid synthase LtaSa (YfnI), the transcription termination factor Rho, endonuclease YazA, and the glucose permease PtsG (Fig. 2b). Transformation experiments confirmed that the mTn insertions were linked to the increased CEF resistance phenotype. The Δ*cpgA* mutant displayed a three-fold increase in the diameter of the zone-of growth inhibition with CEF (42 mm vs. 14 mm for WT; Fig. 2b), which correlates well with the 10-fold difference in MIC considering that the diffusion zone (area) increases as the square of the diameter. This phenotype was complemented by reintroduction of *cpgA* using pMUTIN4[22]. Of the strains carrying suppressor mutations, the greatest level of suppression was observed for the Δ*cpgA* *ptsG::mTn* strain. These results suggest that Δ*cpgA* cells may be sensitized to CEF due to the import of glucose through the PtsGHI (PTS) system. This is surprising, since LB medium generally has low levels of glucose, estimated as <0.1 mM[23].

Next, we monitored the sensitivity of a Δ*cpgA* mutant to glucose. Unexpectedly, Δ*cpgA* was unable to grow in the presence of glucose as monitored by a zone-of-inhibition assay on Mueller-Hinton (MH) medium, a rich medium with abundant amino acids that support gluconeogenic growth (Fig. 2c). This is an unusual phenotype, and we are not aware of any other mutants that are intoxicated by glucose in *B. subtilis*. Null mutations in other neighboring genes (*prpC, prkC, rpe,* and *thiN*) did not show this phenotype (Supplementary Fig. 2), and it could be fully complemented by reintroduction of *cpgA* at locus (Fig. 2c).

To determine if Δ*cpgA* cells are altered more generally in central carbon metabolism we evaluated growth in modified Spizizen minimal medium (MSMM) supplemented with different carbon sources (PTS sugars, PTS-independent sugars, and organic acids). Lack of *cpgA* led to an inability to grow on glucose, gluconate, and ribose (indicated with red circled "X" in Fig. 1), which all feed carbon into the PPP. With other tested carbon sources, including both sugars (fructose, sucrose, mannose, mannitol, N-acetyl glucosamine) and TCA cycle intermediates (malate, fumarate, succinate, citrate), Δ*cpgA* cells

grew to high cell density albeit with an increased lag time relative to WT (indicated with green circled check-marks in Fig. 1, Supplementary Fig. 3).

Like glucose, gluconate impaired the growth of the Δ*cpgA* mutant on otherwise rich medium. Indeed, the Δ*cpgA* mutant was exquisitely sensitive to gluconate with a zone of clearance around the disc (~42 mm diameter) three-fold greater than that for glucose at similar concentration (Fig. 2c), whereas WT and complemented cells were not at all growth inhibited (Fig. 2c). Since *B. subtilis* can use both glucose and malate as preferred carbon sources, and these compounds can be co-metabolized[24], we next asked whether the presence of glucose would interfere with the ability of a Δ*cpgA* mutant to grow using malate as carbon source. As expected Δ*cpgA* cells grew well with malate (Supplementary Fig. 3i), but very poorly when glucose was also present (Fig. 2d). In contrast, glucose toxicity can be partially suppressed by supplementation with fructose (which enters glycolysis as fructose-1,6-bisphosphate) (Supplementary Fig. 3p). Growth was also noted with sucrose, which generates both glucose-6-phosphate and fructose-6-phosphate (Supplementary Fig. 3f).

To determine whether exposure to glucose and gluconate was exerting a bacteriostatic or a bactericidal effect, we used a spot dilution assay to monitor colony-forming units (CFU) after shift of Δ*cpgA* cells from permissive medium (LB) to defined medium containing glucose or gluconate. A shift to high glucose leads to bacteriostasis (Fig. 2e). Even after many hours of exposure to glucose there was no loss in CFU. However, as expected, the Δ*cpgA* cells formed much smaller colonies on LB agar. In contrast, WT and the Δ*cpgA* mutant complemented by re-introduction of CpgA at-locus grew in the presence of glucose with an increase in CFU over time. Similarly, the WT and complemented strain also continued to grow, as evidenced by an increase in CFUs, after gluconate addition (Fig. 2f). However, the Δ*cpgA* cells ceased growth after gluconate addition (no increase in CFUs after 1 h), and had a severely reduced plating efficiency at 7 h (Fig. 2f). In liquid medium, the density (OD$_{600}$) of Δ*cpgA* cells decreases drastically in the presence of gluconate, suggestive of a bactericidal effect. Collectively, these observations suggest that the Δ*cpgA* mutant is intoxicated by a product arising from the PPP downstream of gluconate.

**Reduced Zwf activity suppresses ΔcpgA glucose sensitivity**. To gain insight into the molecular basis for the observed glucose intoxication, we isolated spontaneous suppressors from within the zone of clearance when Δ*cpgA* was plated on MH medium with glucose (Fig. 3a). Suppressor mutations were mapped by whole genome re-sequencing, and four of seven strains contained single nucleotide polymorphisms (SNPs) in *zwf*, encoding glucose-6-phosphate dehydrogenase (G6PDH), which catalyzes the rate-limiting step for entry of carbon into the PPP (Supplementary Table 1; Fig. 3a). Mapping of these mutations onto the G6PDH structure from the Gram-positive bacterium *Leuconostoc mesenteroides* indicates that they are not located in the substrate-binding sites, but likely affect the dimerization interface (Fig. 3b). Suppression was not observed for a Δ*cpgA* *zwf::erm* double mutant (Fig. 3c), suggesting that these alleles were not likely to be null mutations. Suppression was also not observed when a second, IPTG-inducible copy of *zwf* was induced in the Δ*cpgA* background (Fig. 3c). We thus hypothesized that these suppressor alleles might encode altered or reduced function G6PDH.

Next, we tested the ability of a Δ*cpgA* *zwf* double mutant expressing various *zwf* alleles to grow on glucose. When the only copy of *zwf* was present as an IPTG-inducible copy at *amyE*

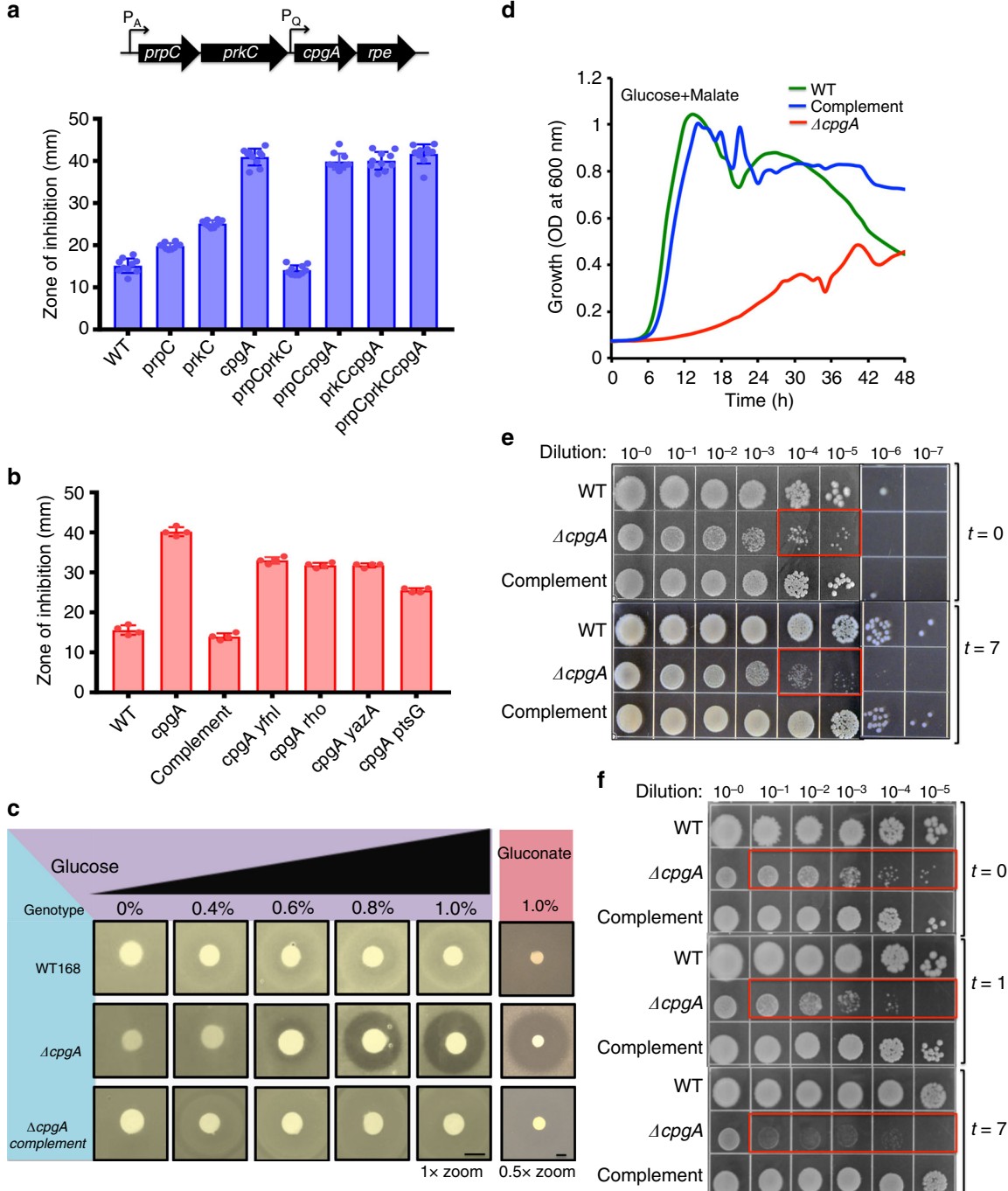

**Fig. 2** A Δ*cpgA* mutant is intoxicated by glucose or gluconate. **a** Top panel: *prpC-prkC-cpgA* genomic locus organization and bottom panel: CEF sensitivity of the strains shown as measured for 6 μg of CEF using a disk diffusion assay (8 mm diameter) performed on LB media (*n* = 10). **b** The CEF sensitivity (shown as zone of inhibition, ZOI, mm) of the Δ*cpgA* strain can be partially suppressed by deletions in genes identified by mTn insertions (*yfnI*, *rho*, *yazA*, and *ptsG*) (*n* = 3). The complemented strain (Δ*cpgA* pMUTIN4-*cpgA*) was at locus. Results shown are representative of three-independent biological replicates. **c** Disk diffusion assays to measure glucose and gluconate sensitivity were determined on MH medium. Under our conditions, 1% glucose (10 mg) is 55 μmols and 1% gluconate is 46 μmols on the filter. **d** Growth (OD$_{600}$ vs time) for various strains in defined MSMM supplemented with 0.6%(w/v) glucose plus 0.6%(w/v) malate. Growth measurements were recorded at every 15 min. for WT (green line), Δ*cpgA* (red line) and complement (blue line) cultures growing aerobically at 37 °C for 48 h. **e** Spot dilution experiments (10 μl) were used to measure cell viability as a function of time after cells growing in MSMM at 37 °C were subjected to 4%(w/v) glucose. **f** Spot dilution experiments (10 μl) were used to measure cell viability after exposure to 4% gluconate (as for panel **e**). These experiments are representative of three-independent biological replicates; all images were captured following 18 h of incubation at 37 °C. Significant observations related to CFU fitness are indicated with red boxes in **e** and **f**

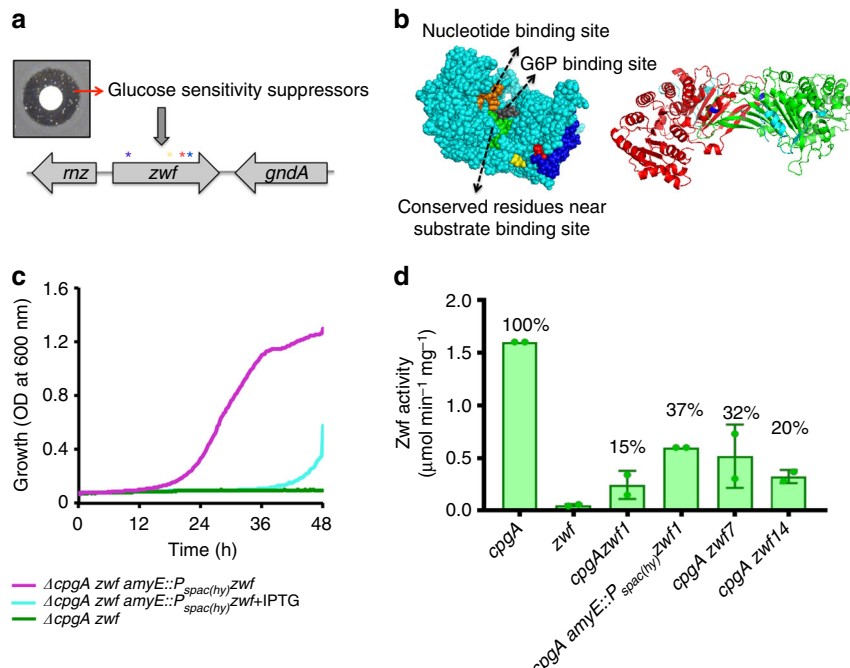

**Fig. 3** Suppression of Δ*cpgA* glucose sensitivity by reduction in Zwf activity. **a** Δ*cpgA* suppressor mutants were recovered (after ≥24 h) from the zone of clearance generated by glucose on MH medium at 37 °C. Four independent mutations in *zwf* are shown (see Supplementary Table 1). **b** Left panel: *B. subtilis* Zwf was modeled (I-TASSER) to generate a space-fill view (cyan) highlighting the nucleotide binding site (orange), glucose-6-phosphate binding site (gray), and conserved residues (green). Suppressor mutations affect D449N (yellow), W455R (red), W455stop (truncation of C terminus loop shown in blue). Right panel: suppressor mutations led to changes in Zwf dimer (red and green protomers) as modeled on the *Leuconostoc mesenteroides* (PDB: 1DPG) structure (45% identical to *B. subtilis* Zwf). Suppressor mutations affected V191L (blue) at the dimer interphase and W455stop (truncation shown in cyan). **c** Growth of a Δ*cpgA*Δ*zwf* double deletion (green line) in MSMM with 0.8% glucose is rescued by *amyE*::P$_{spac(hy)}$*zwf* when uninduced (low level of Zwf, magenta line), but not when Zwf is induced with IPTG (cyan line). **d** Specific activity of Zwf in Δ*cpgA* suppressor strains monitored in cell extracts. NADPH formation (A$_{340}$ vs. time) as a result of G6P oxidation by Zwf was measured and is shown as the percentage relative to the Δ*cpgA* parent strain (n = 2, 2 biological replicates with each consisting of three technical replicates)

(*amyE*::P$_{spac(hy)}$-*zwf*), the Δ*cpgA* strain grew on glucose when the construct was not induced, consistent with a small amount of leaky expression from this promoter, but not when 1 mM IPTG was present (Fig. 3c). This suggests that a low level of G6PDH was needed to support growth, but that higher levels led to glucose-intoxication. In contrast, when the Δ*cpgA zwf* double mutant cells were expressing the mutant *zwf* alleles, growth was observed with 1 mM IPTG (Supplementary Fig. 4). This is consistent with the notion that these suppressor strains contain hypomorphic *zwf* alleles encoding reduced activity G6PDH. Indeed, when Zwf activity was assayed in the Δ*cpgA zwf1*, Δ*cpgA zwf7*, and Δ*cpgA zwf14* strains, there was an ~85%, 68%, and 80% reduction relative to the Δ*cpgA* parent strain, respectively (Fig. 3d). We conclude that glucose intoxication in the Δ*cpgA* strain is relieved by reduction of carbon flux into the PPP.

We further observed that the *zwf* mutant alleles recovered in this selection are dominant. For example, ectopic induction of *zwf1* in a Δ*cpgA* strain, carrying a wild-type *zwf* allele at locus, lead to a ~63% reduction in Zwf activity (Fig. 3d). Moreover, induction of wild-type Zwf cannot rescue the poor growth of Δ*cpgA* strain on glucose, whereas induction of several hypomorphic alleles does (Supplementary Fig. 4). Since G6PDH is a dimer, we hypothesized that heterodimer formation might reduce Zwf activity in the merodiploid cells. Mutations in the human Zwf ortholog (G6PDH) give rise to G6PDH deficiency, an X chromosome-linked disease that primarily affects males. These mutations are also often found at the dimer interface and lead to a reduction of enzyme activity[25]. It is not clear whether any of these mutant alleles are dominant since there is stochastic X-chromosome inactivation in females[26], and therefore the phenotype of cells expressing both WT and mutant alleles is not obvious.

**6-phosphogluconate intoxicates Δ*cpgA* cells**. Spontaneous suppressors were also recovered from Δ*cpgA* cells challenged with gluconate (Fig. 4a). In this case, whole genome re-sequencing mapped a suppressor mutation to *gntP*, a permease involved in gluconate uptake (Fig. 4a; Supplementary Table 1). Similarly, a Δ*cpgA gntK* double mutant (lacking the gluconokinase responsible for converting D-gluconate to 6-phosphogluconate; Fig. 1) was also tolerant of gluconate (Fig. 4a). These results indicate that gluconate toxicity requires import and phosphorylation. As predicted, mutations in *zwf* that suppress glucose toxicity did not suppress gluconate toxicity, which enters the PPP downstream of Zwf (Fig. 1).

To test whether growth inhibition was due to accumulation of 6-phosphogluconate or a downstream metabolite, we overexpressed GndA within Δ*cpgA* cells. Induction of either GndA (the major, NADP-dependent GndA) or the gluconate-inducible paralog, GntZ, restored growth of the Δ*cpgA* mutant strain on modified Spizizen's minimal medium (MSMM) containing glucose (Fig. 4b, c). We conclude that 6-phosphogluconate, rather than a downstream metabolite, is toxic to cells. Overexpression of GndA also partially suppressed the CEF sensitivity of Δ*cpgA*, suggesting that CEF sensitivity results, at least in part, from dysregulation of central carbon metabolism (Supplementary Fig. 5).

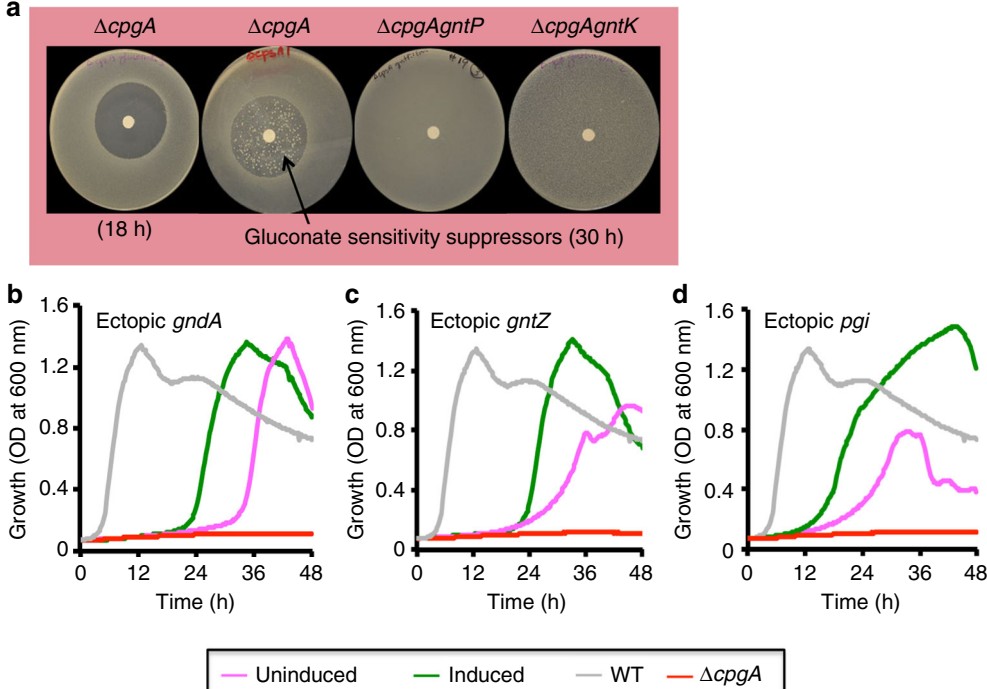

**Fig. 4** Intoxication of Δ*cpgA* cells by 6PG, a potent inhibitor of PGI. **a** Images of disk diffusion assays performed in MH media with ~20 μmols gluconate for various strains. Arrow indicate colony formation within the zone of clearance, indicative of suppressor mutations, as seen for Δ*cpgA* upon extended incubation (30 h). Gluconate resistance is observed for the Δ*cpgA gntP* and Δ*cpgA gntK* strains. Growth curves are shown for WT (gray line), Δ*cpgA* (red line), and Δ*cpgA* containing ectopic constructs at *amyE* locus for **b** *gndA* **c** *gntZ* and **d** *pgi*. Ectopic strain constructs were tested with (*gndA* induced with 250 nM, *gntZ* with 0.5 mM, and *pgi* with 1 mM IPTG, green line) and without IPTG (pink line) in MSMM with 0.8% glucose. Aerobic growth was monitored every hour for the period of 48 h at 37 °C. The data set is representative of three or more independent biological replicates

**6-phosphogluconate inhibits PGI**. 6-phosphogluconate has been widely studied as a competitive inhibitor of phosphoglucoisomerase (PGI), the initiating enzyme that directs glucose-6-phosphate into glycolysis. PGI is constitutively expressed and is amongst the most abundant proteins during growth on glucose[27,28] (Fig. 1). Previously, 6-phosphogluconate was found to be a potent inhibitor of PGI from *Bacillus caldotenax* with a $K_i$ of 30 μM[29]. Consistent with the hypothesis that 6-phosphogluconate can inhibit PGI, we found that overproduction of PGI restored the ability of Δ*cpgA* cells to grow on MSMM with glucose as a sole carbon source (Fig. 4d), with an even greater reduction in lag time than noted for overexpression of GndA or GntZ (Fig. 4b, c). In contrast, overproduction of several other enzymes of central carbon metabolism (including Zwf, YkgB, Rpe, FbaA, GapA, Pgm, Pgk, and Eno) did not restore growth on glucose (Supplementary Fig. 4). We conclude that the proximal cause of growth arrest in Δ*cpgA* cells is a failure of PGI due to inhibition by 6-phosphogluconate generated from the PPP. This can also account for the ability of Δ*cpgA* cells to grow with sucrose (Supplementary Fig. 3f), which simultaneously provides cells with both glucose-6-phosphate and fructose-6-phosphate, thereby bypassing the need for PGI (Fig. 1).

**4PE inhibits GndA**. Next, we searched for known inhibitors of GndA activity. In both yeast and mammals, it was recently noted that 4PE is a potent inhibitor of GndA that is degraded by a broad specificity phosphatase[30,31]. This phosphatase, known as Pho13 (4-nitrophenyl phosphatase) in yeast and PGP (phosphoglycolate phosphatase) in mammals, dephosphorylates 4PE. 4PE arises when glyceraldehyde-3-phosphate dehydrogenase (GAPDH) uses the PPP intermediate erythrose-4-phosphate as an accidental substrate[30]. Although CpgA is not homologous to Pho13/PGP,

this led us to hypothesize that CpgA may function as a phosphatase to cleanse the metabolite pool of potentially toxic inhibitors.

To determine if the growth defects of the Δ*cpgA* strain might be related to 4PE accumulation and consequent inhibition of GndA, we expressed the yeast Pho13 phosphatase in Δ*cpgA* cells. Induction of Pho13 (with 50 μM IPTG) protected Δ*cpgA* cells against gluconate-mediated killing (Fig. 5a) and allowed growth in MSMM supplemented with glucose (Fig. 5b). In general, Δ*cpgA* cells are slow growing even in LB medium, and Pho13 expression restored growth comparable to that of WT. Since Pho13 is specific for the hydrolysis of 4PE[30], these results strongly suggest that 4PE is responsible for GndA inhibition in Δ*cpgA* cells.

**CpgA functions as a metabolite-proofreading phosphatase**. To determine if CpgA functions as a metabolite-proofreading phosphatase, we used LC-MS to compare intracellular metabolite levels in WT and Δ*cpgA* cells 30 min. after shift from LB to MSMM containing 0.8% glucose. Strikingly, loss of *cpgA* led to at least a 100-fold increase in intracellular 4PE, which was below the detection limit in WT cells (Fig. 5c, d). In addition, we noted a more modest increase in intracellular accumulation of 6-phosphogluconate (2.2-fold), as well as GTP (1.7-fold) and erythrose-4-P (1.4-fold). This compares with an 8-fold increase in 6-phosphogluconate in yeast lacking Pho13[30]. In contrast, other phosphorylated intermediates were largely unchanged including hexose-1-phosphates (the assay used did not distinguish glucose-1-phosphate and fructose-1-phosphate). Thus, CpgA may function analogously to Pho13 to dephosphorylate 4PE, and perhaps other metabolites, and thereby prevent inhibition of GndA.

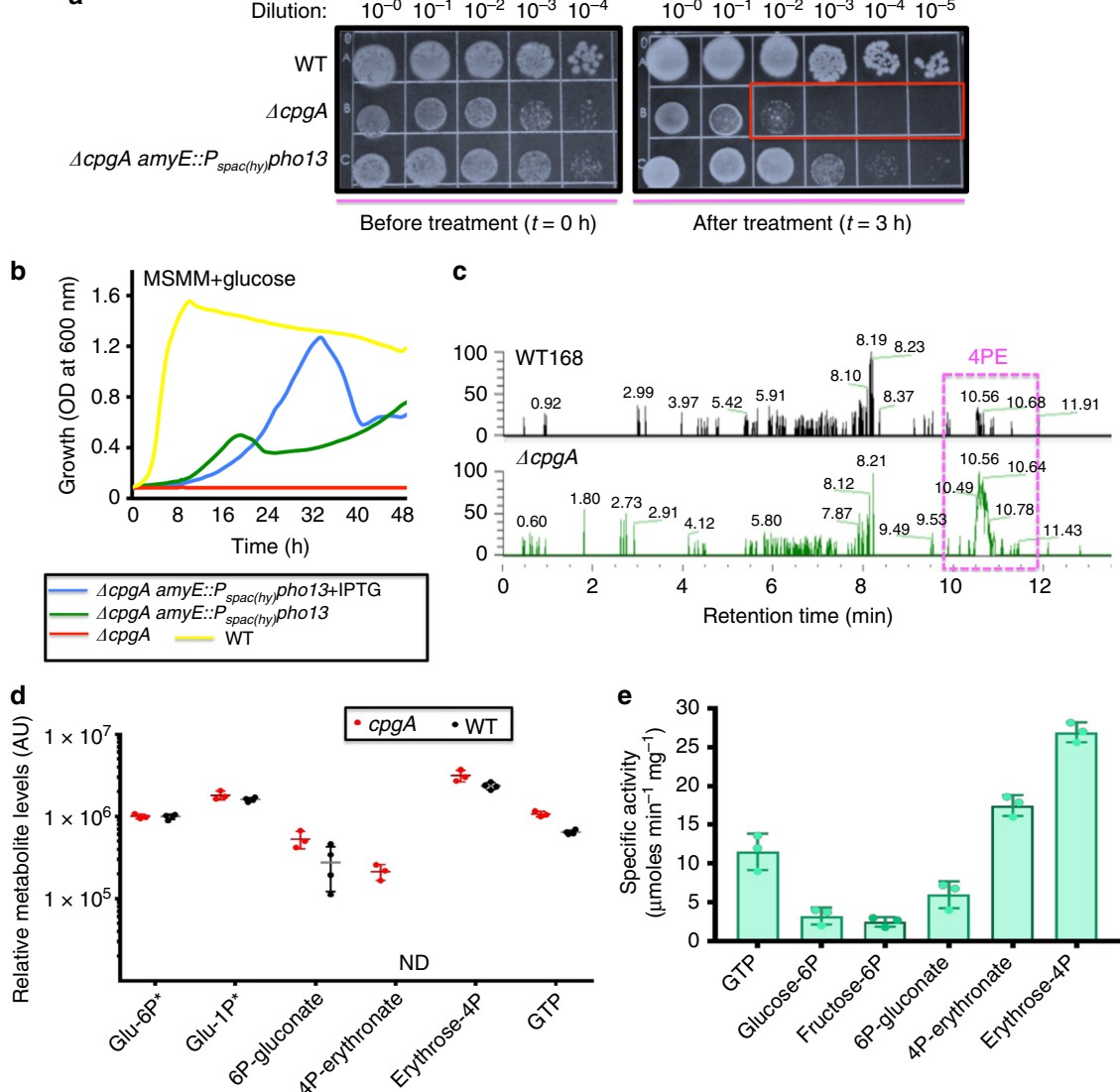

**Fig. 5** Inhibition of 6PG dehydrogenase is due to 4PE. **a** Spot dilutions performed on LB medium to determine cell viability (CFUs) after exposure of cells growing in MSMM to 4%(w/v) gluconate for 0 min. (left panel) to 3 h (right panel) illustrating the >2 log reduction (red box) in Δ*cpgA* vs. Δ*cpgA* expressing the Pho13 phosphatase. **b** Growth monitored in MSMM with 0.8%(w/v) glucose is rescued in Δ*cpgA* cells ectopically expressing Pho13 with (blue line) and without 0.1 mM IPTG (green line). Growth trend of WT (yellow line) and Δ*cpgA* (red line) strains are also included for comparison. **c** Extracted Ion Chromatogram (EIC) for WT and Δ*cpgA* cells for mass of 4PE eluted from retention time of 0–12 min. **d** Intracellular metabolite levels (mean ± standard error of mean of normalized ion counts for $n = 4$ of independent biological replicates for mutant group, and $n = 3$ biological independent replicates for WT group) in WT and Δ*cpgA* cells as measured 30 min after shift of cells into MSMM medium with 0.8%(w/v) glucose. ND represents non-detectable levels of 4PE in WT. Origin for y-axis is set at $10^4$ representing levels at least 10-fold above the detection limit. **e** In vitro phosphatase activity of purified CpgA with different substrates was measured using a malachite green assay ($n = 3$)

To explore the substrate range of CpgA, we purified the protein and tested a variety of candidate substrates (GTP, G6P, F6P, 6-phosphogluconate, erythrose-4-phosphate, and 4PE) (Fig. 5d). The highest levels of phosphatase activity were observed for erythrose derivatives: erythrose-4-phosphate and 4PE (~25–28 μmole min$^{-1}$ mg$^{-1}$ of CpgA). This is nearly identical to the rate measured with Pho13 using 4PE (~30 μmole min$^{-1}$ mg$^{-1}$), although Pho13 had very low activity with erythrose-4-phosphate[30]. GTP and 6-phosphogluconate were also moderately good substrates for CpgA (~10–12 μmole min$^{-1}$ mg$^{-1}$). Much lower rates were observed for G6P and F6P (~2–3 μmole min$^{-1}$ mg$^{-1}$). Previously, a K177A mutation the Walker A motif of CpgA was shown to abolish GTPase activity[6]. Consistent with a central role for the CpgA active site in metabolite proofreading, expression

of a CpgA(K177A) mutant did not allow growth on glucose (Supplementary Fig. 3a). These data strongly suggest that CpgA is a broad specificity phosphatase with a preference for erythrose-derived phosphosugars.

These results support a model in which the CpgA protein functions as a metabolite repair phosphatase with activity against the accidental metabolite 4PE. By analogy with the findings in yeast and mammals, 4PE is likely generated by the essential enzyme GapA[30]. In the absence of CpgA, 4PE triggers an inhibition cascade in which 4PE binding to GndA leads to elevated levels of 6-phosphogluconate, a potent inhibitor of PGI. The coordinated inhibition of both glycolysis (PGI) and the PPP (GndA) creates metabolic gridlock, thereby precluding growth in the presence of glycolytic and PPP-dependent carbon sources.

**CpgA metabolite proofreading is independent of PrkC.** CpgA is encoded as part of a complex operon including the *prpC-prkC-cpgA* triad of genes (Fig. 2a), an organization conserved in the *Firmicutes*. Moreover, CpgA has been shown to be phosphorylated by PrkC on Thr166[6], and this modification increases GTPase activity and association with 70S ribosomes[6]. These findings suggest that PrkC may regulate one or more activities of CpgA. To determine whether or not PrkC-mediated phosphorylation affects the metabolite-proofreading role of CpgA, we replaced *cpgA* with alleles encoding phosphoablative (CpgA T166A) or phosphomimetic (CpgA T166E) CpgA variants. Strikingly, cells expressing any of the three CpgA variants grew similarly in either LB medium (Supplementary Fig. 6a) or MSMM plus glucose (Supplementary Fig. 6b). In addition, they were equally resistant to CEF as monitored by either disk diffusion (Supplementary Fig. 6c) or liquid culture MIC determinations (Supplementary Fig. 7). Our microscopic observations are consistent with those previously reported[10,32], with a subset of cells displaying obvious morphological and cell division defects (Supplementary Fig. 6d). These morphology defects were not affected by phosphorylation status of Thr166 in our investigation.

These results indicate that CpgA phosphatase activity does not require PrkC-dependent phosphorylation, consistent with the lack of glucose sensitivity for a Δ*prkC* mutant (Supplementary Fig. 2).

**CpgA functions in two distinct pathways.** Cells lacking CpgA are slow growing even in rich medium (LB, MH, MH + 0.4% fructose), and this defect is exacerbated at lower temperatures. Cold-sensitivity is characteristic of ribosome-assembly defects[33], consistent with the assigned role of CpgA as an RA-GTPase[2]. However, our results suggest that CpgA additionally functions as a metabolite proofreading phosphatase. To determine if these are two distinct functions, or if the reported defects in ribosome assembly might also be related to metabolite intoxication, we tested whether Δ*cpgA* strains overexpressing either PGI or Pho13 would overcome the cold-sensitivity of parental Δ*cpgA* strain on LB agar. Although induction of PGI or Pho13 significantly increased the fitness of Δ*cpgA* cells at 37 °C (Fig. 6a), their expression was unable to rescue cell growth at 30 °C (Fig. 6b). In contrast, complementation with CpgA, which provides both the ribosome assembly and metabolite proofreading

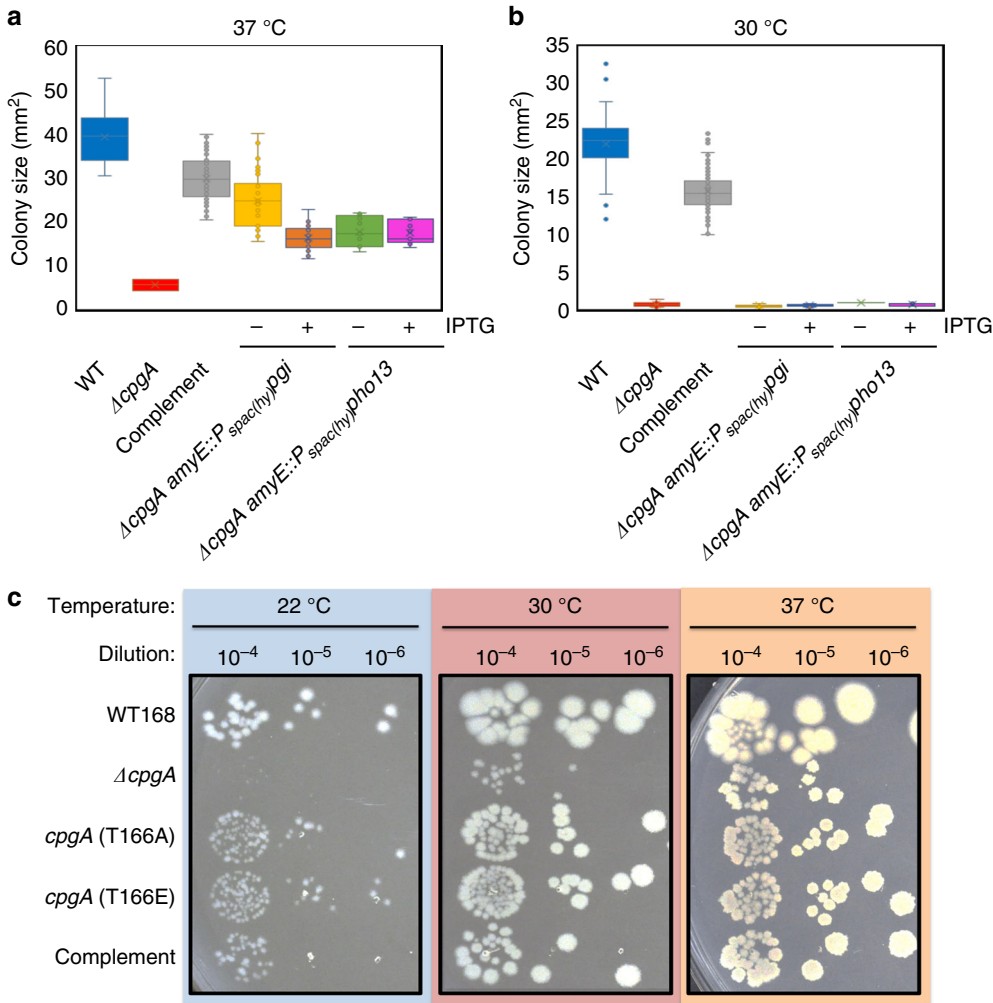

**Fig. 6** CpgA has separable functions in ribosome assembly and metabolite proofreading. Box plot of colony area (*n* ≥ 45) determined with ImageJ software for various strains (WT in blue, Δ*cpgA* in red, complement in gray, Δ*cpgA* expressing Pgi with (orange) and without IPTG (yellow), and Δ*cpgA* expressing Pho13 with (pink) and without IPTG (green) plated at 37 °C (**a**) and at 30 °C (**b**) for 40 h. **c** All strains were grown to OD$_{600}$ of ~0.4, serially diluted, and spotted onto LB agar plates with incubation at 22 °C, 30 °C, or 37 °C for 40 h. Boxplot depicting centerline represents the second quartile (median) and the bottom and top of the box are the first and third quartiles, respectively and the X inside the box is the mean. Whiskers extend 1.0 quartile range. Outliers are shown as single dots

functions, fully rescues growth at both temperatures. CpgA, as well as its phosphomimetic and phosphoablative variants, were equivalent in their ability to support growth at a variety of temperatures (Fig. 6c). Although PrkC-dependent phosphorylation might influence the activity of CpgA in ribosome assembly and metabolite proofreading, we have not found conditions where this impact is apparent.

Our analyses support the notion that CpgA is a bifunctional protein, with activities in both ribosome assembly and metabolite proofreading, where the relative importance of these two activities depends on growth conditions. Our genetic suppression studies suggest that both functions may contribute to fitness in the presence of β-lactam antibiotics under the conditions tested. In our original selection for CEF resistance on LB medium, the strongest effect was found for *ptsG::mTn*, which likely reduced metabolic stress elicited by import of glucose. However, we also recovered an insertion just upstream of *yazA*, encoding a predicted endonuclease. The *yazA* gene is co-transcribed with *yabB* (encoding a tRNA methyltransferase) and *yabC* (encoding a likely 16 S rRNA methyltransferase). We speculate that alterations in the expression of one or more of these genes may partially compensate for the lack of the ribosome assembly function of CpgA.

In previous work, strong genetic interactions were observed between *rsgA* (the *E. coli cpgA* ortholog) and a variety of genes implicated in ribosome assembly[34]. We postulated that mutations that compensate for the ribosome assembly defect of ΔcpgA cells might be selected if the metabolite proofreading function was provided by the heterologous metabolite-proofreading enzyme Pho13. To test this idea, we selected for suppressor mutations that increased growth of ΔcpgA on medium containing glucose plus gluconate, but with induction of Pho13. In this condition, several of the recovered mutations affected ribosome related functions (e.g. ribosome recycling factor, rRNA methyltransferase) (Supplementary Table 1). Thus, under conditions conducive to 4PE formation (glucose plus gluconate and loss of CpgA), Pho13 can compensate for metabolite intoxication, and we propose that the resulting suppressor mutations in this strain are instead compensating for defects in ribosome assembly.

## Discussion

CpgA is one of 17 P-loop type GTPases encoded in *B. subtilis*, 12 of which are thought to have functions related to translation [translation factor-related, or TRAFAC, GTPases[35]]. A subset of these, including CpgA, are specifically involved in ribosome-assembly and have been designated as RA-GTPases[2]. Previous genetic studies have revealed that ΔcpgA mutants have a range of phenotypes, many of which are not obviously explained by a defect in ribosome assembly. For example, ΔcpgA mutants are growth compromised, and the severity of the growth defect is dependent on media conditions. Indeed, *cpgA* was even assigned as an essential gene in early studies[9]. In addition to the observed growth defects, ΔcpgA mutants are sensitive to a range of antibiotics including both those targeting the ribosome and some cell wall synthesis inhibitors[5]. While sensitivity to ribosome-targeting antibiotics might be expected for cells with a defect in ribosome assembly, the basis for the sensitivity to cell wall inhibitors is less obvious. Cells depleted for (or lacking) CpgA also display altered cell morphology[6].

Here, using forward genetics, we ascribe an unexpected function to CpgA as a broad specificity, metabolite proofreading phosphatase. This function is essential during growth on carbon sources that feed directly into upper glycolysis or the PPP. Indeed, cells lacking CpgA are intoxicated by growth on glucose or gluconate, even when other preferred carbon sources are present

(Fig. 2). We ascribe this intoxication to an inhibition cascade in which accidentally generated 4PE inhibits the primary GndA and this leads to an elevation in cellular levels of 6-phosphogluconate, which inhibits the glycolytic enzyme PGI. The simultaneous inhibition of both glycolysis (PGI) and the PPP (GndA) leads to metabolic gridlock and eventually cell death. Evidence in support of this model includes (i) a drastic (>100×) increase in 4PE in *cpgA* mutant cells after shift to glycolytic carbon sources (Fig. 5c), (ii) suppression of toxicity by overproduction of either GndA, the GndA paralog GntZ, or PGI (Fig. 4b–d), (iii) suppression of toxicity by yeast Pho13, an enzyme that specifically catabolizes 4PE (Fig. 5a, b), and (iv) biochemical evidence that CpgA is a broad specificity phosphatase with 4PE as a preferred substrate (Fig. 5d).

Although *B. subtilis* uses a wide range of organic compounds as sources of carbon and energy, glucose and malate are preferred carbon sources. During growth on glucose, a significant fraction (~43%) of G6P is diverted to the PPP, while during growth on malate as little as ~8% of flux is allocated to the PPP loop[24]. In cells lacking CpgA, high flux through the PPP loop during growth on glucose or gluconate results in growth inhibition, even when other preferred carbon sources are present. Our results suggest that the proximal cause of growth inhibition is the loss of PGI activity, since overproduction of PGI effectively suppresses glucose or gluconate toxicity (Fig. 4). Consistently, cells grow comparatively well on sucrose (Supplementary Fig. 3f), which provides both glucose-6-phosphate and fructose-6-phosphate, thereby compensating for the inhibition of PGI in the ΔcpgA mutant (Supplementary Fig. 3f; Fig. 1). Interference with PGI activity may also be related to the observed CEF sensitivity in ΔcpgA cells. Indeed, we have previously shown that availability of F6P, a branch point intermediate critical for the production of aminosugars used in peptidoglycan synthesis, can affect CEF sensitivity[20].

The generation of 4PE as a toxic metabolite is an example of what is often termed underground metabolism. The enzymes of glycolysis and the PPP are among the most abundant in the cytosol[36], and it is increasingly appreciated that these and other metabolic enzymes may sometimes function with alternative substrates. In some cases, these secondary activities may be physiologically beneficial[37,38]. However, in other cases the products of these adventitious reactions may be toxic or may represent metabolic dead-ends, in which case they may be considered as accidents of metabolism[30,39]. Recently, the PPP intermediate erythrose-4-P was found to adventitiously react with the promiscuous enzyme GAPDH to generate 4PE, an inhibitor of GndA. Mammals contain a phosphoglycolate phosphatase capable of hydrolyzing 4PE, and in yeast this activity was associated with the ortholog Pho13[30,31]. These phosphatases thus function as metabolite proofreading enzymes that destroy toxic compounds generated during glycolytic growth. Here, we provide evidence that CpgA has an analogous function.

In *E. coli* 4PE is not an accidental metabolite, but a precursor used to support the synthesis of vitamin B6 by the deoxyxylulose-5-phosphate (DXP) pathway[40]. In this pathway, 4PE is synthesized by erythrose 4-P dehydrogenase (Epd), a GAPDH homolog. In contrast, *B. subtilis* synthesizes vitamin B6 by a short pathway involving two enzymes PdxST, and without formation of 4PE[40]. Rosenberg et al.[41] engineered a *B. subtilis* strain lacking *pdxST* to express the last two enzymes of the DXP pathway (PdxH and PdxJ), and were then able to evolve strains that could grow in the absence of PLP. Their findings imply that, at least in their evolved strains, *B. subtilis* produces sufficient 4PE to support the function of the engineered DXP pathway[41].

In conclusion, we propose that CpgA is a bifunctional protein with roles in both ribosome assembly and metabolite

proofreading. It is increasingly appreciated that proteins may have two unrelated functions, a phenomenon referred to as protein moonlighting[42]. For example, aconitase functions as an enzyme in the tricarboxylic acid cycle, and also as an RNA-binding protein to regulate gene expression[42]. For CpgA, the relative importance of its two activities depends on both the growth medium and temperature. The metabolite-proofreading role of CpgA is essential for growth of *B. subtilis* in the presence of glucose or gluconate. In contrast, the role in ribosome assembly seems to be most critical for growth at low temperatures where ribosome assembly is potentially compromised. These findings highlight the perils of assigning protein functions solely based on homology, and serve as a reminder that proteins may participate in multiple, independent pathways.

## Methods

**Bacterial strains, growth conditions, and media**. The strains used in this study are listed in Supplementary Table 2. The primary strains for gene deletions were obtained from the BKE/BKK collection of disruptants containing erythromycin/kanamycin cassettes[21] available from the *Bacillus* Genetic Stock Center (BGSC). Mutations were moved into *B. subtilis* 168 and then the antibiotic marker removed using pDR244 plasmid to generate clean, in-frame deletions as described[21]. For ectopic expression, genes were amplified with primers listed in Supplementary Table 3 from *B. subtilis* subspecies 168 chromosomal DNA using Phusion high-fidelity DNA polymerase (NEBL) and subjected to restriction enzyme digestion, purification, and ligation into pre-digested pPL82 vector for propagation in *E. coli* DH5α on LB medium (Affymetrix) supplemented with 100 µg ml⁻¹ of ampicillin. Prepared recombinant plasmid constructs were transformed into Δ*cpgA* for insertion at the *amyE* locus by double-cross over recombination selected with 10 µg ml⁻¹ chloramphenicol. For cis, at-locus complementation Δ*cpgA* was amplified with 600 bp flanking regions (for homology) and cloned into pMUTIN4 plasmid using conventional restriction enzyme cloning. The recombinant construct was propagated in *E. coli* and transformed into Δ*cpgA* for single cross over selected with 1 µg ml⁻¹ of erythromycin. Long-flanking homology PCR fragments containing desired phospho-replacement mutations were generated and purified prior to transformation into Δ*cpgA* strains with selection with 1 µg ml⁻¹ of erythromycin. All strains were verified by junction-PCR and sequencing. For all of the growth experiments, cultures were streaked from frozen glycerol stocks onto Luria-Bertani (LB) agar plates grown at 37 °C for 18 h. Cells were grown in 5 ml of LB broth, supplemented when required with antibiotics (1 µg ml⁻¹ erythromycin and/or kanamycin 15 µg ml⁻¹ and/or 10 µg ml⁻¹ chloramphenicol), at 37 °C under shaking conditions of 300 rpm on a gyratory shaker. Once reaching OD₆₀₀ ~0.4, cells were harvested by centrifugation, washed multiple times in buffered saline, pH 7.4, and resuspended in MSMM to be used as a starter culture. MSMM was dispensed in Honeycomb 2 multi-well plates (Steri) at OD₆₀₀ of 0.07 supplemented with appropriate antibiotics and IPTG when needed. Growth in MSMM supplemented with various carbon sources (0.8–1% glucose, 0.8% gluconate, 0.6% malate, 0.8% fructose, 0.8% sucrose, 0.8% fumarate, 0.8% mannose, 0.8% ribose) or 0–0.1 µg of CEF (sigma) was monitored for up to 48 h at 37 °C under shaking conditions using Bioscreen C growth curve analyzer (Growth Curves USA, NJ).

**Disk diffusion assay**. CEF sensitivity was evaluated in solid LB agar medium using a zone of inhibition assay[43]. The bottom agar contained 15 ml LB broth agar (with 1.5% final agar concentrations), and the top soft-agar consisted of 4 ml LB broth agar (with 0.75% agar final agar concentrations) and 100 µl of mid-exponentially grown cultures (A₆₀₀ₙₘ of 0.4) as inoculum. Whatman filter paper disks (8 mm in diameter) with CEF (6 µg) were placed on the plates and the zone of growth inhibition (ZOI) was measured after 18 h at 37 °C. For carbon source related sensitivities, we employed a similar approach involving a range of glucose and gluconate concentrations tested on Mueller Hinton broth (Fluka analytical) agar.

**Mariner-transposon (mTn) mutagenesis**. The Δ*cpgA* cells were transformed with pMarA plasmid[44] and selected on LB medium containing erythromycin (1 µg ml⁻¹) at 30 °C giving rise to HB20429. HB20429 cells were grown under similar aerobic conditions to maintain and propagate pMarA at 30 °C until OD₆₀₀ reached ~0.8 and cells were plated onto selection media (LB with CEF prepared by agar poured and solidified at a slanting angle or MSMM agar containing glucose as a carbon source) at 42 °C. Colonies that grew after overnight incubation were sub-cultured in LB supplemented with kanamycin (15 µg ml⁻¹) at 37 °C to confirm the presence of mTn. Genomic DNA was prepared and purified from these cells and was subjected to TaqaI restriction enzyme digestion at 37 °C for 2 h, followed by overnight ligation of cohesive ends to generate a circular transposon-gDNA chimeric library. The PCR reaction from ends of the mTn performed on chimeric library was further subjected to sequencing analysis to identify the orientation and location of mTn insertion.

**Spot dilutions and survival curves**. Overnight cultures grown on LB agar plates were used to inoculate 5 ml liquid LB with aerobic growth at 37 °C until OD₆₀₀ ~0.4, then cultures were centrifuged, washed multiple times with phosphate buffered saline, (pH 7.4). The washed cells were then re-suspended into 5 ml of MSMM containing 4% glucose, (pH 7.4) or 5 ml of MSMM containing 4% of sodium gluconate, (pH 7.4). 100 µl of samples was collected at various time intervals (t = 0, 1, 2, 3, 4, 5, 6, and 7 h) during incubation at 37 °C under 300 rpm of shaking conditions. All of the harvested samples were subjected to serial dilutions (10⁻⁰ to 10⁻⁸) using PBS, and 10 µl of diluent along with undiluted samples were spotted onto freshly prepared, square-grid LB agar plates (25 ml). Spotted cultures were air-dried in a laminar-hood for 15 min. and plates were initially grown at 30 °C for 6 h and then were shifted to 37 °C for further growth for 6 h and observations were made by counting colony forming units at the end of the incubation period. For phospho-replacement study strains were grown to OD₆₀₀ of 0.4 and were serially diluted and 10 µl of cultures were spotted and air-dried for 15 min. in laminar air flow and plates were incubated at 22 °C, 30 °C, and 37 °C for 40 h and images were capture using standard camera.

**Whole genome re-sequencing**. Spontaneous suppressor mutants were picked and their phenotype confirmed. Chromosomal DNA was extracted and purified using Qiagen genomic DNA extraction kit. Genomic DNA was quantified by Qubit™ dsDNA HS assay kit (ThermoFisher scientific) and 50 ng of DNA samples were submitted to Biotechnology Resources at Cornell University core facility (Ithaca, NY). DNA sequences determined using Illumina HiSeq2500 (with 75 bp paired end reads) were analyzed using CLC genomic workbench (Qiagen) to trim, map, and align against the *B. subtilis* 168 reference genome (Ref Seq: NC_000964.3) to detect SNPs. Results were confirmed using Sanger DNA sequencing.

**Zwf enzyme assay**. G6P dehydrogenase (Zwf) enzyme activity was measured from crude extracts[45]. Extracts were prepared from 5 ml of cell culture at OD₆₀₀ of ~0.8 harvested and suspended in 1 ml 20 mM Tris buffer (pH 7.4), and subjected to sonication for 60 sec. Briefly, 0.25 mM NADP⁺, 2.5 mM of MgCl₂, and 50 µl of crude bacterial extract in 20 mM diglycine buffer, pH 9.4 were mixed. The reaction was initiated with 12.5 mM G6P (Sigma) and absorbance at 340 nm was monitored before and after addition using a Synergy H1 plate reader (BioTek Instruments, Inc. VT) as a function of time.

**Phosphate (Pi) release assay**. N-terminal histidine-tagged CpgA was purified as described by Absalon and coworkers[10]. Then, His-tagged CpgA was dialyzed in 20 mM HEPES (pH 8) buffer, and protein concentration was determined using Bradford colorimetric assay kit. In a 96 well plate, Pi release was measured using 10 µM CpgA with 1 mM of substrates: guanosine-5'-triphosphate (Roche), D-glucose-6-phosphate (Sigma), D-fructose-6-phosphate (Sigma), D-gluconate-6-phosphate (Sigma), D-erythrose-4-phosphate (Sigma), and D-erythronate-4-phosphate (Santa Cruz Biotechnology), in a 100 mM MOPS buffer (pH 7.5) containing 1 mM MgCl₂. The reactions were incubated at 37 °C for 20 min. and released phosphates were measured spectroscopically at 620 nm by adding 25 µl of malachite green working reagent (Sigma-Aldrich, MAK308) incubated for 30 min at room temperature.

**Metabolite extraction**. For metabolite measurements, cell cultures of *cpgA* mutant and its isogenic *B. subtilis* 168 WT strains were grown aerobically in LB media to OD of 0.6 at 37 °C. In total, 40 ml was centrifuged and subjected to several washing steps using PBS and cells were resuspended in MSMM containing 0.8% glucose, and cells were further incubated at 37 °C for 30 min. Cells were vacuum filtered, and the filter was dropped into quenching solution of 60% methanol containing 0.9% ammonium bicarbonate held at −40 °C. Cells were gently scraped and washed-off of membrane filters and subjected to rinse using 0.9% ammonium bicarbonate (4 °C). Recovered cells were then transferred to 70% methanol solution pre-cooled to −40 °C. The ultra-sonication of cells in organic solvent was performed on ice (at power 30 W) for 5 min and cell debris was subjected to centrifugation at 10,000 × *g* for 15 min at 4 °C. The supernatant was collected and vacuum dried and was stored at −80 °C until further use.

**Metabolite detection**. Samples were re-suspended with 150 µl 60% ACN prior to LC/MS analysis. Chromatographic separation was performed on a Vanquish UHPLC system with a SeQuant ZIC pHILIC column (5 µm, 2.1 × 150 mm) coupled to a Q Exactive™ Hybrid Quadrupole-Orbitrap High Resolution Mass Spectrometer (Thermo Fisher Scientific, San Jose, CA, USA). The mobile phase consisted of (A) 10 mM ammonium acetate in water (pH 9.8), 0.1% formic acid and (B) acetonitrile. The gradient was as follows: 0–15 min, 90–30% solvent B; 15–18 min, isocratic 30% solvent B; 18–19 min, 30–90% solvent B; 19–27 min, 90% solvent B; followed by 3 min of re-equilibration of the column before the next run. The flow rate was 250 µl min⁻¹ and the Injection volumes were set to 2 µl.

All of the samples were analyzed by negative electrospray ionization (ESI) in full scan MS mode. Nitrogen as sheath, auxiliary, and sweep gas was set at 50, 8, and 1 U, respectively. Other conditions included the following: resolution, 120,000 full width at half maximum; automatic gain control target, 3 × 10⁶ ions; maximum injection time, 100 ms; scan range, 67–1000 m z⁻¹ (mass to charge ratio); spray

voltage, 3.50 kV; and capillary temperature, 275 °C. ESI data-dependent MS-MS spectra were generated for quality control pool samples, for identification purposes, through the use of the following conditions: resolution, 15,000 full width at half maximum; automatic gain control target, $10^5$ ions; maximum injection time, 50 ms; isolation window, 0.4 $mz^{-1}$ (mass to charge ratio); and normalized collision energy of 20, 30, 40. The acquired data set, composed of full MS and data-dependent MS-MS raw files, was processed using Compound Discoverer 2.1. Initial untargeted metabolomics workflow with putative identification through ChemSpider and mzCloud etc., databases were used for processing the raw data. The software parameters for alignment were 5 ppm mass tolerance for the adaptive curve model and 0.5 min maximum shift for alignment. The software parameters for detecting unknown compounds were 5 ppm mass tolerance for detection, 30% intensity tolerance, 3 for the sensitivity and noise threshold, and $2 \times 10^6$ minimum peak height. Pure compounds were used as standards to calculate the retention time for specific compounds in targeted approach.

**Colony size measurements**. Bacterial cultures were picked from fresh LB agar plates containing appropriate antibiotics and grown at 37 °C to OD$_{600}$ ~0.4 and serially diluted. Hundred microliters of cultures were spread on LB agar plates with and without IPTG using aseptic glass beads and plates were incubated at various temperatures (30 and 37 °C) and images were captured post 40 h of incubation. Images were processed using Fiji-ImageJ software to calculate the area of each colony.

**Reporting summary**. Further information on experimental design is available in the Nature Research Reporting Summary linked to this article.

## Data availability
A reporting summary for this Article is available as a Supplemental Information file. All data supporting the findings of this study are available from the corresponding author upon request.

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

## Acknowledgements

This work was supported by National Institutes of Health R35GM122461 awarded to JDH. We are thankful to Diana Herrera, Gumpanat Mahipant, Bacillus Genetic Stock Center, and SubtiWiki[46] database for support. We acknowledge Biotechnology Resources at Cornell (BRC) metabolomics facility for help with metabolite measurements, and Dr. Frédérique Pompeo (CNRS-Laboratoire de Chimie Bactérienne) for the gift of strains harboring pOMG360 construct, which was originally constructed by the Séror lab.

## Author contributions

Conception, A.J.S. and J.D.H.; Designed and performed experiments, A.J.S.; Manuscript drafted and edited, A.J.S. and J.D.H.

## Additional information

**Competing interests:** The authors declare no competing interests.

