## [Peer Review File · Nature Communications]

Reviewers' comments:

Reviewer #1 (Remarks to the Author):

Sachla and Helmann with report about a second function of the ribosome assembly GTPase in the disposal of a toxic byproduct of metabolism, 4-phosphoerythronate (4PE). The authors provide compelling evidence that toxicity of glucose and gluconate for a *cpgA* mutant result from the accumulation of 4PE, and they demonstrate in vitro that CpgA indeed has phosphatase activity for this metabolite. It is becoming increasingly clear that the formation of toxic byproducts of metabolites and their disposal is a very important field of research that is just at its emergence. This study makes an important contribution!

Specific comments:

#1 Would a *gndA gntZ* double mutation rescue the *cpgA* mutant?

#2 p. 7: Why does the deletion of *zwf* not suppress glucose toxicity to the *cpgA* mutant?

#3 l. 162: What are "hypomorphic alleles"?

#4 ref. 37: provide the complete information (volume, page numbers)

Jörg Stülke

Reviewer #2 (Remarks to the Author):

This is a very well-written manuscript presenting genetically elegant and exhaustive analysis of the conditional growth defect of a *cpgA* mutant of *B. subtilis* in the presence of glucose and some other carbon sources. The experiments lead to an important and exciting discovery of a novel metabolic proofreading function for CpgA, apparently unrelated to its known function in ribosome assembly.

1. My only major comment concerns the conclusion that CpgA has two separate functions. It is likely correct, but formally one may argue that overexpression of PGI or Pho13 (p. 11) does not fully suppress the metabolic defect and this residual metabolic defect is sufficient for the defect in ribosome assembly (alternatively, the ribosome assembly defect is irrelevant and it is the residual metabolic defect that gets exacerbated at low temperatures). To make their case stronger, the authors could state (or determine) whether the ribosome assembly defect/cold-sensitivity exists in a *cpgA* mutant under other growth conditions that do not confer an apparent metabolic defect, e.g., in the presence of fructose or glycerol (i.e., when 4PE is not produced or is not toxic).

Very minor and editing comments:

2. Please, comment on the difference between the bacteriostatic effect of glucose and the bacteriocidal effect of gluconate.

3. line 195: please, change "GndA" to "overproduction of GndA".

4. lines 197-198: see lines 185-187.

5. lines 205-207: please, focus this sentence on the inhibitor and not on the phosphatase.

6. line 209: "glyceraldehyde".

7. line 244: please, indicate that GapA is essential.

8. lines 342, 345, 347: please, avoid restatements and explain the difference between "accidental" and "underground". Why should "accidental" metabolism be necessarily detrimental?

9. line 354: see lines 220-223. The authors' own data prove this point better.

10. lines 355-361: it is not clear what the authors are trying to say (and multiple references are required for their statements). Also, see line 244 for a possible contradiction to (or confusion with) line 355.

11. lines 437, 478: please, correct the references.

Reviewer #3 (Remarks to the Author):

In this genetic tour de force the Helmann group shows that a widely conserved GTPase (named CpgA in *Bacillus subtilis*) involved in ribosome assembly is also a phosphatase involved in metabolite repair. It hydrolyses 4-phosphoerythronate (4PE) that is known toxic side intermediate of the pentose phosphate pathway.

The study is extremely well conducted, rigorous, controlled and logically written. It was a real pleasure to read, every time I was thinking to myself "oh, they should have checked this", it was done in the following section. The Helmann lab uses classical and modern bacterial genetics techniques combined with deep knowledge of bacterial and *Bacillus subtilis* physiology. To start just with a phenotype, e.g. sensitivity to B-lactams and unravel the whole molecular chain of events is actually difficult to do. This can become a classical example.

I think the study is interesting for a broader audience beyond the ribosome assembly and metabolite repair communities because it is an example of a protein with two bona-fide functions in two totally different areas of the cellular machinery. These cases are not unprecedented but rare.

I would suggest that the authors add a paragraph in the discussion listing a few other such cases. Also the only missing logical piece in the study is that the authors do not really go back to explaining their initial b-lactam sensitivity phenotype. This should be discussed.

We thank the Editor for the prompt handling of our manuscript and all three referees for the supportive and enthusiastic comments

Our responses to referee comments follow (responses with **):

REFEREE 1

#1 Would a *gndA gntZ* double mutation rescue the *cpgA* mutant?

** Actually, we show that overproduction (not loss) of GndA (or its paralog GndZ) rescues the mutant. In a double mutant, loss of both 6-phosphogluconate dehydrogenases (GndA/GntZ) causes buildup of 6-phosphogluconate (6-PG) which blocks phosphoglucose isomerase (Pgi), hence it does not rescue loss of *cpgA*.

#2 p. 7: Why does the deletion of *zwf* not suppress glucose toxicity to the *cpgA* mutant?

** The PPP pathway is important for generating nucleic acid sugars, vitamins, and reducing equivalents. Under conditions of *cpgA*, our genetic results indicate that a reduction (but not loss) of *Zwf* is advantageous.

#3 l. 162: What are “hypomorphic alleles”?

** Hypomorphic refers to a partial loss of gene function, as in the case of *zwf* alleles with reduced activity that suppress a *cpgA* null. This part was rewritten so that the definition of hypomorphic is now clear from the context where it first appears.

#4 ref. 37: provide the complete information (volume, page numbers)

** The full reference is shown: Trends Biotechnol 37, 29-37 (2018)

REFEREE 2

1. My only major comment concerns the conclusion that CpgA has two separate functions. It is likely correct, but formally one may argue that overexpression of PGI or Pho13 (p. 11) does not fully suppress the metabolic defect and this residual metabolic defect is sufficient for the defect in ribosome assembly (alternatively, the ribosome assembly defect is irrelevant and it is the residual metabolic defect that gets exacerbated at low temperatures). To make their case stronger, the authors could state (or determine) whether the ribosome assembly defect/cold-sensitivity exists in a *cpgA* mutant under other growth conditions that do not confer an apparent metabolic defect, e.g., in the presence of fructose or glycerol (i.e., when 4PE is not produced or is not toxic).

** We performed experiment as prescribed by reviewer where permissive medium Mueller Hinton agar and Mueller Hinton agar supplemented with 0.4% Fructose could not overcome cold-sensitivity of *cpgA* mutants.

Very minor and editing comments:

2. Please, comment on the difference between the bacteriostatic effect of glucose and the bacteriocidal effect of gluconate.

** This section was rewritten to more clearly explain the relevant observations in Fig. 2e vs. 2f. In addition, we have added text to indicate that "In liquid medium, the density (OD₆₀₀) of Δ cpgA cells decreases drastically in the presence of gluconate, suggestive of a bactericidal effect" (line 144).

3. line 195: please, change "GndA" to "overproduction of GndA".

** Corrected as suggested

4. lines 197-198: see lines 185-187.

** We corrected the inadvertent redundant text.

5. lines 205-207: please, focus this sentence on the inhibitor and not on the phosphatase.

** Sentence reordered for clarity

6. line 209: "glyceraldehyde".

** corrected

7. line 244: please, indicate that GapA is essential.

** updated as requested.

8. lines 342, 345, 347: please, avoid restatements and explain the difference between "accidental" and "underground". Why should "accidental" metabolism be necessarily detrimental?

** We have removed the redundancy and refer to "underground metabolism", which is the most widely used term in the field. We explain that in some cases underground metabolism can lead to toxic or dead-end products, which is a type of metabolic accident, as noted in the cited reference, and necessitates the presence of metabolite proofreading activities.

9. line 354: see lines 220-223. The authors' own data prove this point better.

** Agreed. We have rewritten and condensed this section to be clearer in our intent, which is simply to provide a supporting example of how 4PE, an accident of metabolism, may actually allow the evolution of a new pathway (underground metabolism).

10. lines 355-361: it is not clear what the authors are trying to say (and multiple references are required for their statements).

** We have simplified the text and added references as noted.

Also, see line 244 for a possible contradiction to (or confusion with) line 355.

11. lines 437, 478: please, correct the references.

** We have simplified the text and corrected references were needed.

REFEREE 3

I think the study is interesting for a broader audience beyond the ribosome assembly and metabolite repair communities because it is an example of a protein with two bona-fide functions in two totally different areas of the cellular machinery. These cases are not unprecedented but rare. I would suggest that the authors add a paragraph in the discussion listing a few other such cases. Also the only missing logical piece in the study is that the authors do not really go back to explaining their initial β -lactam sensitivity phenotype. This should be discussed.

** We have added a reference to the concept of protein moonlighting, and more clearly defined the dual roles of CpgA in this context. We have also added a sentence to the Discussion to address the issue of CEF sensitivity: "Interference with PGI activity may also be related to the observed CEF sensitivity in $\Delta cpgA$ cells. Indeed, we have previously shown that growth conditions can influence the availability of F6P, a branch point intermediate critical for the production of aminosugars used in peptidoglycan synthesis, and that this can affect CEF sensitivity²⁰."